# AI-generated text boundary detection with RoFT

**Laida Kushnareva**[1], **Tatiana Gaintseva**[2],
German Magai[3,4], Serguei Barannikov[5,6], Dmitry Abulkhanov, Kristian Kuznetsov[5],
Eduard Tulchinskii[5], Irina Piontkovskaya[1], Sergey Nikolenko[7]

[1] AI Foundation and Algorithm Lab, Russia;
[2] Digital Environment Research Institute, Queen Mary University of London, UK;
[3] HSE University, Russia; [4] Noeon Research, Japan;
[5] Skolkovo Institute of Science and Technology, Russia;
[6] CNRS, Université Paris Cité, France;
[7] St. Petersburg Department of the Steklov Institute of Mathematics, Russia

**Correspondence:** kushnareva.laida@gmail.com

## Abstract

Due to the rapid development of large language models, people increasingly often encounter texts that may start as written by a human but continue as machine-generated. Detecting the boundary between human-written and machine-generated parts of such texts is a challenging problem that has not received much attention in literature. We attempt to bridge this gap and examine several ways to adapt state of the art artificial text detection classifiers to the boundary detection setting. We push all detectors to their limits, using the Real or Fake text benchmark that contains short texts on several topics and includes generations of various language models. We use this diversity to deeply examine the robustness of all detectors in cross-domain and cross-model settings to provide baselines and insights for future research. In particular, we find that perplexity-based approaches to boundary detection tend to be more robust to peculiarities of domain-specific data than supervised fine-tuning of the RoBERTa model; we also find which features of the text confuse boundary detection algorithms and negatively influence their performance in cross-domain settings.

## 1 Introduction

Artificial text detection (ATD) is a very difficult problem in real life, where machine-generated text may be intertwined with human-written text, lightly edited, or pad out human-generated prompts. However, in literature ATD is usually formulated in a simpler way, with a dataset of text samples labeled as either entirely human-written or entirely machine-written, so the detection problem can be safely treated as binary classification. Moreover, the models for this binary classification are often developed and trained to detect a particular type of generator, e.g., text produced by a specific large language model (LLM) (Uchendu et al. (2023)). This is in stark contrast with how we may encounter artificially created text in real life, where documents partially written by humans and partially generated by LLMs already abound. This setting is much more complex and much less researched.

In this work, we experiment with a lesser known dataset called RoFT (Real Or Fake Text), collected by Dugan et al. (2023). Each text in this dataset consists of ten sentences, where the first several sentences are human-written and the rest are machine-generated starting from this prompt, mainly by models from the GPT family (Radford et al., 2019; Brown et al., 2020). We consider several techniques developed for binary ATD, modifying them for this more complex boundary detection setting; e.g., following Tulchinskii et al. (2023a) we adapt intrinsic dimension estimation which is currently considered to be the most robust method

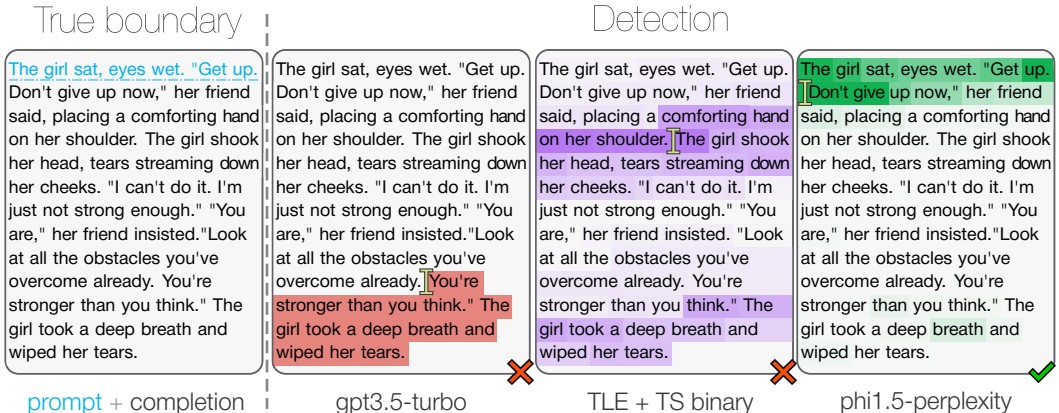

Figure 1: Sample input from the ROFT-chatgpt dataset. The rightmost part of the picture shows the true bondary between human-written and machine-generated text. Other three parts are colored according to the features, extracted by various classifiers. Here, gpt3.5 predicts that only two last sentences are generated; TLE + TS binary predicts that generation starts from the second sentence; phi1.5 perplexity predicts that generation starts from the third sentence which is correct answer. See Section 3 for detailed explanation of all the methods and Appendix G for more examples from the dataset.

for cross-domain and cross-model binary detection. Fig. 1 shows a sample input from RoFT and results of the considered models. Our primary contributions are as follows: (1) we evaluate five approaches to adapt perplexity-based detectors to the task of detecting the boundary between human-written and machine-generated text, discussing the differences in their behavior compared to binary ATD and providing a comprehensive analysis of how perplexity scores react to the machine–human transition in the text; we also compare several backbones for the perplexity evaluation to show which model works the best for this purpose; (2) we present evidence that perplexity scores obtained from small language model, trained on data generated by bigger LLM, provide strong features for boundary detection models, leading to relatively high accuracy of detection and cross-domain robustness related to other detection methods; (3) we introduce and evaluate two ways to adapt the classifiers based on intrinsic dimension estimation for this task, showing how the time series analysis can be used for extracting useful information from Transformer representations; (4) we show how the robustness of boundary detectors (including the fully-tuned RoBERTa baseline) to domain shift depends on the particular properties of the domains; we study the properties of the dataset itself and their effect on the performance of every approach; (5) we enrich the RoFT dataset with *GPT-3.5-turbo*[1] (ChatGPT) generation samples; we share this new dataset with the community, establish baselines, and analyze the behavior of our detectors on it.[2]

We hope that this work will encourage further research both in boundary detection for texts that are partially human-written and partially generated and analyzing how inner representations of Transformer-based models react to such transitions; the latter direction may also help interpretability research. The rest of the paper is organized as follows. In Section 2 we survey related work, and Section 3 introduces the methods we have applied for artificial text boundary detection. Section 4 presents a comprehensive evaluation study on the RoFT and RoFT-chatgpt datasets. Section 5 presents a detailed discussion and analysis of our experimental results, and Section 6 concludes the paper.

## 2 Related Work

**Artificial text detection** (ATD) is a well-studied task with plenty of already considered approaches; see, e.g., the surveys by Yang et al. (2023); Uchendu et al. (2023); Wu et al.

---

[1]https://platform.openai.com/docs/model-index-for-researchers
[2]Our official repository: https://github.com/SilverSolver/ai_boundary_detection.git

(2023a). In our work, we mostly concentrate on adapting two groups of methods to artificial text boundary detection: methods based on perplexity estimation since they are the most widely used and methods based on topological data analysis (TDA) since they have shown promising robustness to domain shift and model shift (see below).*Perplexity-based methods* are based on the idea that human-written text tends to be more unpredictable and varied compared to AI-generated text which may follow more predictable patterns, i.e., has a lower perplexity score; among them we note GPTZero (Weber-Wulff et al., 2023), Sniffer (Li et al., 2023a), SeqXGPT (Wang et al., 2023), and LLMDet (Wu et al., 2023b). *Topological data analysis* (TDA) for ATD is inspired by the results of Kushnareva et al. (2021) and Tulchinskii et al. (2023a). The latter has shown that for many artificial text generators the intrinsic dimensions (PHD) of RoBERTa embeddings of the texts created by these generators are typically smaller than those of the texts written by humans; also TDA has proven to be useful for closely related tasks of fake news detection (Tudoreanu), authorship attribution (Elyasi & Moghadam, 2019), and detection of synthetic speech (Tulchinskii et al., 2023b).

**Style change detection and authorship attribution**. ATD can be considered as a special case of authorship attribution (AA), where the LLM is one author and the human is another, and the task is to determine the authorship of different parts of the text; it is also similar to the style change detection problem in multi-author documents (Zangerle et al., 2021). Jones et al. (2022) show that LLM models can successfully imitate human style and deceive existing popular online AA methods. However, Venkatraman et al. (2023) propose an approach based on the principle of uniform information density that can detect the authorship of LLM. State of the art style change detectors (Lin et al., 2022; Jiang et al., 2022; Lao et al., 2022; Iyer & Vosoughi, 2020) are based on Transformer-based encoders such as BERT Devlin et al. (2019), RoBERTa Liu et al. (2019), AlBERT Lan et al. (2020), and ELECTRA Clark et al. (2020). Therefore, we use RoBERTa model as a baseline and a source of embeddings.

**RoFT (Real Or Fake Text)**. Our main dataset (Dugan et al., 2020; 2023) originates from a website called RoFT[3] developed as a tool to analyze how *humans* detect generated text, including an extended study of whether and how they can explain their choice, when they say that a text sample is machine-generated, and how humans can learn to recognize machine-generated text better. Every user of this website can choose a topic ("Short Stories", "Recipes", "New York Times", or "Presidential Speeches") and start the following "game". The player sees ten sentences one by one. The first sentence is always written by a human, but for each subsequent sentence the player must determine whether it is machine-generated or human-written. If the player believes the text was AI-generated, they should explain why they think so. If they guess machine generation before the true boundary, they earn zero points. Otherwise, they earn $5 - x$ points, where $x$ is the number of sentences after the correct boundary. This is actually a harder problem than just boundary detection since the player does not see the full text, and the scoring function is skewed. As a result, Dugan et al. (2023) created a dataset also known as RoFT, where every sample consists of ten sentences, starting with a human-written prompt and continuing by some language model; the data shows the true boundary, human prediction of it, the generator model, an explanation provided by the player, and information about the player. The original RoFT contains generations from GPT-2 (Radford et al., 2019), GPT-2 XL, GPT-2 finetuned on the "Recipes" domain, GPT-3.5 (davinci), CTRL (Keskar et al., 2019) with control code "nocode" and with control code "Politics", and baseline, where instead of an LLM-generated continuation the passage transitions to a completely different news article selected at random. RoFT has been mostly used to investigate how humans detect artificially generated texts manually (Clark et al., 2021); Cutler et al. (2021) provided the first baselines on automatically solving RoFT, comparing several shallow classification and regression models based on RoBERTa and SRoBERTa (Reimers & Gurevych, 2019) embeddings collected from the last layer of these models. However, the cross-domain and cross-model settings in their research were very limited. In fact, Cutler et al. (2021) call "Out of Domain" (OOD) classifiers trained on *all* available data and then evaluated on a given subset, and they call "In Domain" (ID) classifiers that had been trained and evaluated on the same subset. Besides, they did not analyze all generators and domains within this cross-domain setting. In this work,

---

[3]http://www.roft.io/

we concentrate on cross-domain and cross-model settings and interpretability, evaluating boundary detectors on unseen generators (models) and topics (domains).

**Other related work**. Zeng et al. (2023) used a TriBERT-based approach for artificial text boundary detection in student essays for educational purposes, but with no cross-domain problem setting. Wang et al. (2023) address mixed human and AI generations by sentence-level classification, but they do it with a *white-box detector* that needs access to the generator model, which severely limits applicability. Recently, a SemEval workshop[4] introduced a small artificial text boundary detection dataset, namely subtask C of Task 8[5]. The subtask asks to predict the word where human text ends and artificial text begins. However, to this moment there is no information on (a) which model was used to generate the samples, (b) what are the particular domains or topics of the generations, and (c) any human baselines. Existing technical reports on the competition (Spiegel & Macko, 2024; Datta et al., 2024; Rad et al., 2024) concentrate on other subtasks of Task 8 such as binary ATD and LLM-authorship attribution, but not the boundary detection subtask.

## 3 Approach

We consider several different approaches, including a multilabel classification framework, as proposed by Cutler et al. (2021), where the label of a text corresponds to the number of the first generated sentence, time series analysis that slides a window over the text tokens, and regression methods that minimize the MSE between true and predicted boundaries. We design our classifiers based on features that have been successfully used in prior works on ATD (Solaiman et al., 2019; Mitchell et al., 2023). We also introduce a new baseline based on sentence lengths. Below, we consider all of these approaches in detail.

**RoBERTa classifier**. Unlike Cutler et al. (2021) who process each sentence separately, we fine-tune the RoBERTa model (Liu et al., 2019) to represent the entire text sample via the [CLS] vector. This is the only case in our work where we apply full model fine-tuning. For other methods, we use simpler classifiers such as logistic regression (LR) or gradient boosting (GB) (Friedman, 2001) trained on various features extracted directly from larger models (e.g., Transformer-based LLMs), with no updates to the larger model's weights.

**Perplexity from generative LLMs**. In our perplexity experiments, we implement the *black-box approach*, which means that a single model is used for all data to compute sentence-wise perplexity. This is more practical than the method of Cutler et al. (2021) who used perplexity scores from the original generator model, which may be infeasible if the generator is unknown, in particular in cross-model scenarios. Even for known generators, the exact *version* of the model is usually unknown if they are available via API, which harms the performance of *white-box* detectors. To derive perplexity features, we calculate the log likelihood of each token via LLM, and derive sentence-level *mean* and *std* values. As the perplexity estimator, we use several models such as GPT-2 (Radford et al., 2019), Phi-1 (Gunasekar et al., 2023), Phi-1.5 (Li et al., 2023b), Phi-2 (Abdin et al., 2023), and LLaMA-2-7B (Touvron et al., 2023). Our underlying hypothesis here is that texts generated by models of similar architecture such as GPT-2 or GPT-3.5 might appear more "natural" to these models, as reflected by their likelihood scores. Our findings corroborate this assumption (see Appendix F). On top of sentence-wise perplexity features, we train either a *classifier* (logistic regression and gradient boosting) or a *regressor* that predicts boundary values (GB regressor); the regression setting takes advantage of the sequential nature of the task, minimizing the discrepancy in labels.

**DetectGPT**. The DetectGPT framework (Mitchell et al., 2023) proposes a more nuanced perplexity-based scoring function. It involves perturbing a text passage and comparing log-probabilities between original and altered texts. This score then serves as an input to a classification model, with GPT-2 as the base model and T5-Large (Raffel et al., 2020) generating the perturbations.

---

[4]https://aclanthology.org/venues/semeval/
[5]https://github.com/mbzuai-nlp/SemEval2024-task8

**Length-Based Baseline**. Since we have observed a statistical difference in sentence length distributions between human-written and generated texts (see Fig. 2), we leverage sentence lengths as a simple baseline feature. This baseline allows us to gauge the effectiveness of a classifier in identifying boundaries without semantic understanding.

**Topological Time Series (TTS)**. Inspired by Tulchinskii et al. (2023a), we explore the potential of topological features based on intrinsic dimensionality (ID); we provide an introduction to topological data analysis (TDA), including the definitions of features, in Appendix A. We hypothesize that geometric variations in token sequences can help identify AI-generated text, so we introduce models that process TDA-based features treating them as time series. For every text, we slide a window of $H = 100$ tokens (step size $S = 1$) over RoBERTa token embeddings and find the intrinsic dimension (PHD) of the points within the window, as shown in Fig. 4 (Schweinhart, 2020). Window size 100 was chosen as a minimal size where PHD estimator stayed stable. The time series are then classified with a multi-label SVM with the global alignment kernel (GAK) (Cuturi, 2011), where label correspond to a number of the first generated sentence; this method is called "PHD + TS ML" in the tables.

**Topological binary classification**. In this approach, we train a binary classifier to distinguish between *fake* and *natural* text atop a specific predictor. For the base predictor, we employ intrinsic dimensionality calculated over a sliding window of $H = 20$ tokens (step size $S = 5$)[6]. The TLE (tight local) intrinsic dimension estimator was chosen due to its robust performance on small data samples; window size 20 was taken directly from Amsaleg et al. (2019), as the authors claim that their algorithm is stable enough "for 'tight' localities consisting of as few as 20 sample points". For the base classifier, we use gradient boosting trees (Friedman, 2000; 2002). To translate probabilities predicted with a binary classifier into a specific boundary that separates real and fake text, we determine the final label by maximizing $\mathbf{y} \mapsto \arg\max_{I \in \mathcal{I}} s_I(\mathbf{y}, \mathbf{x})$, where the score function $s_I$ is defined as $s_I(\mathbf{y}, \mathbf{x}) = -\sum_j^n I_{\mathbf{y}}^j \log \hat{p}(\mathbf{x}^j)$, and $\hat{p}(\mathbf{x})$ are predictions from the base binary classifier. In this context, we use a binary indicator set $I \in \mathbf{1}^n$ consisting of a chunk of $k$ zeros followed by another chunk of $n - k$ ones. This method is called "TLE + TS Binary" in the tables.

**Zero-shot detection**. As an additional baseline, we tried zero-shot boundary detection with two strongest black-box models: gpt-3.5-turbo and gpt-4. We describe the problem to them with a natural language prompt which is provided in Appendix B.

## 4   Experimental evaluation

**Dataset preparation and analysis**. In all experiments, the task is to detect the exact boundary where a text passage that started as human-written transits to machine generation. In addition to RoFT (Dugan et al., 2020), we created a new dataset called **RoFT-chatgpt**, where the same human prompts are continued with the *gpt-3.5-turbo* model. RoFT-chatgpt is designed to be more challenging for artificial text boundary detection while preserving the basic statistical properties of RoFT such as the label distribution. We removed duplicates that were different only in human-predicted labels; for *RoFT-chatgpt*, we also removed samples containing "As an AI language model..." and short failed generations. As a result, we retained 8943 samples from the original RoFT and 6940 samples in *RoFT-chatgpt*. Note that distributions of sentence lengths vary significantly across subdomains, as shown in Fig. 2 and Figs. 5, 6 in Appendix E (see also a discussion in Section 5).

**Artificial text boundary detection**. Table 1 presents the main results of our experiments on the RoFT and RoFT-chatgpt datasets; the majority class prediction is the last (9th) class of fully human-written texts. On the original RoFT, we also include the human baseline and the best result reported by Cutler et al. (2021), obtained by a classifier built upon concatenated SRoBERTa embeddings of each sentence. We report the accuracy (*Acc*) and two metrics that account for the sequential nature of boundary detection: soft accuracy, i.e., the percentage of predictions that differ from the correct label by at most one (*SoftAcc1*; indeed, Fig. 18

---

[6]Stride size 5 was chosen empirically after the preliminary experiments. In these preliminary experiments we found out that for smaller stride sizes the series became too long for the Time Series SVM algorithm, and it wasn't able to converge.

Table 1: Boundary detection results. **Bold** shows the best method, underlined - second best, *italic* for values outperforming the human baseline.

| Method | RoFT | | | RoFT-chatgpt | | |
|---|---|---|---|---|---|---|
| | Acc | SoftAcc1 | MSE | Acc | SoftAcc1 | MSE |
| RoBERTa + SEP | **49.64** % | **79.71** % | **02.63** | **54.61** % | **79.03** % | **03.06** |
| RoBERTa | 46.47 % | 74.86 % | 03.00 | 39.01 % | 75.18 % | 03.15 |
| Based on Perplexity | | | | | | |
|    *DetectGPT* + GB classifier | 19.79 % | 37.40 % | *08.35* | 21.69 % | 43.52 % | 06.87 |
|    *DetectGPT* + LR classifier | 19.45 % | 33.82 % | *09.03* | 15.35 % | 41.43 % | 07.22 |
|    *GPT-2 Perpl.* + GB classifier | 24.25 % | 47.23 % | *11.68* | 34.94 % | 59.80 % | 07.46 |
|    *GPT-2 Perpl.* + GB regressor | 12.58 % | 36.67 % | *06.89* | 19.74 % | 54.03 % | 04.89 |
|    *GPT-2 Perpl.* + LR classifier | 23.75 % | 42.15 % | *15.80* | 33.50 % | 57.56 % | 09.25 |
|    *Phi-1 Perpl.* + GB classifier | 20.9 % | 42.7 % | *12.92* | 36.5 % | 56.0 % | 09.13 |
|    *Phi-1 Perpl.* + GB regressor | 10.7 % | 33.0 % | *07.51* | 17.4 % | 49.9 % | 05.51 |
|    *Phi-1 Perpl.* + LR classifier | 19.2 % | 38.3 % | *16.26* | 31.0 % | 53.9 % | 09.76 |
|    *Phi-1.5 Perpl.* + GB classifier | 29.7 % | 55.0 % | *09.50* | 50.8 % | 74.1 % | 04.59 |
|    *Phi-1.5 Perpl.* + GB regressor | 17.1 % | 44.6 % | *06.11* | 32.0 % | 71.0 % | 03.07 |
|    *Phi-1.5 Perpl.* + LR classifier | 27.0 % | 49.5 % | *11.90* | 47.3 % | 72.7 % | 04.77 |
|    *Phi-2 Perpl.* + GB classifier | 35.4 % | 58.3 % | *08.80* | 48.4 % | 70.3 % | 05.39 |
|    *Phi-2 Perpl.* + GB regressor | 15.6 % | 43.7 % | *05.83* | 30.8 % | 68.7 % | 03.33 |
|    *Phi-2 Perpl.* + LR classifier | 25.8 % | 49.5 % | *11.78* | 44.4 % | 71.5 % | 05.42 |
|    *LLaMA-2-7B Perpl.* + GB classifier | 22.3 % | 42.0 % | *14.79* | 41.5 % | 65.0 % | 06.32 |
|    *LLaMA-2-7B Perpl.* + GB regressor | 11.4 % | 32.9 % | *7.98* | 23.6 % | 60.3 % | 03.89 |
|    *LLaMA-2-7B Perpl.* + LR classifier | 19.4 % | 34.7 % | *19.75* | 29.7 % | 58.4 % | 07.90 |
| Based on TDA | | | | | | |
|    PHD + TS ML | *23.50* % | *46.32* % | 14.14 | 17.29 % | 35.81 % | 14.45 |
|    TLE + TS Binary | 12.58 % | *30.41* % | 22.23 | 20.02 % | 34.58 % | 18.52 |
| Zero-shot | | | | | | |
|    gpt-3.5-turbo | 02.5 % | 17.3 % | 30.3 | 07.1 % | 23.9 % | 25.5 |
|    gpt-4 | 06.4 % | 22.0 % | 27.0 | 09.1 % | 25.0 % | 24.1 |
| *Length* + GB | 14.64 % | 33.43 % | 16.55 | 25.72 % | 46.18 % | 18.99 |
| Majority class | 15.26 % | 25.43 % | 27.58 | 13.83 % | 24.42 % | 26.46 |
| SRoBERTa (Cutler et al., 2021) | 42 % | — | — | | — | |
| Human baseline | 22.62 % | 40.31 % | 13.88 | | — | |

in Appendix F shows that many misclassifications point to an adjacent class, which may be acceptable in real world applications), and mean squared error (MSE). More precisely, $SoftAcc1 = \frac{1}{n} \sum_{i=1}^{n} \mathbb{1}_{\{|y_i - \hat{y}_i| \leq 1\}}$ and $MSE = \frac{1}{n} \sum_{i=1}^{n} (y_i - \hat{y}_i)^2$, where $y_i$ - predictions of the algorithm, $\hat{y}_i$ - true labels, $n$ - amount of the examples.

First, on the original *RoFT* RoBERTa-based classifiers outperform others by a large margin (14% accuracy) and also significantly outperform the previously best reported SRoBERTa (Cutler et al., 2021). This model also provides the lowest MSE (0.03) among all methods. We note, however, that our RoBERTa classifier has significantly more trainable parameters than any other method in the table because no other approaches require LM fine-tuning. Second, topological and perplexity features improve over the human baseline. Perplexity-based classifiers are the best in terms of accuracy, while the perplexity regressor provides good MSE values (recall from Section 2, however, that humans were solving a harder problem with a somewhat different objective). Third, RoBERTa's accuracy on *RoFT-chatgpt* drops by 6% compared to RoFT, while soft accuracy and MSE are roughly the same, but the opposite holds for perplexity-based methods: on *RoFT-chatgpt* their results are much *better*. The reason for this might be that we used GPT-like models for perplexity calculation and GPT-3.5-turbo to generate fake samples in *RoFT-chatgpt*. However, we used smaller models to detect text generated by a larger model (GPT-3.5) and got second-best results among other approaches, despite Mitchell et al. (2023) reporting that smaller models are not capable of detecting text generated by larger models. Moreover, when analyzing the performance of different perplexity estimators, we do not observe a clear correlation of the performance with the size of the estimator. The largest LLaMA model with 7B parameters works much worse than Phi-1.5 (1.3B) and Phi-2 (2.7B) on both datasets and slightly worse

Table 2: Accuracy for leave-one-out cross-domain evaluation on *RoFT-chatgpt*. △ and ▽ show relative change from the model's *in-domain* score to the human score; ▲ and ▼, relative change from the *out-of-domain* score to the *in-domain* score. *Green* highlights improvements, *red*, deteriorations. The table shows only perplexity methods based on perplexity of Phi1.5 and Phi2; for other perplexity backbones, see Appendix C.

| Pred. | Model | Context | Pres. Speeches IN ↑ | OUT ↑ | Recipes IN ↑ | OUT ↑ | New York Times IN ↑ | OUT ↑ | Short Stories IN ↑ | OUT ↑ | Avg Δ ↓ |
|---|---|---|---|---|---|---|---|---|---|---|---|
| Text | RoBERTa SEP | global | 57.3△153% | 31.4▼45% | 43.2△91% | 13.1▼70% | 53.2△135% | 38.1▼28% | 54.3△140% | 28.6▼47% | −48% |
| Text | RoBERTa | global | 67.6△199% | 36.3▼46% | 53.7△134% | 14.8▼72% | 54.2△137% | 38.0▼30% | 64.8△183% | 36.1▼44% | −52% |
| Perpl. | **Phi1.5, GB** | sent. | 52.5△132% | **52.0**▼1% | 60.4△167% | 24.1▼60% | 52.5△132% | 45.7▼12% | 48.7△115% | **56.1**△15% | −15% |
| Perpl. | Phi1.5, LR | sent. | 48.7△115% | 41.2▼15% | 60.9△165% | 21.1▼65% | 50.8△125% | 45.2▼11% | 47.1△108% | 51.5▲9% | −21% |
| Perpl. | Phi2, GB | sent. | 50.1△121% | 46.1▼8% | 55.6△146% | 22.8▼41% | 55.0△143% | 42.1▼23% | 48.1△113% | 53.9△12% | −15% |
| Perpl. | Phi2, LR | sent. | 49.6△119% | 42.1▼15% | 56.4△149% | 24.4▼43% | 50.5△123% | 43.4▼14% | 48.5△114% | 52.0▲7% | −16% |
| PHD | TS multi | 100 tkn | 20.3▽10% | 13.7▼32% | 19.2▽15% | 19.5▲02% | 20.9▽08% | 17.2▼18% | 21.2▽06% | 17.6▼17% | −16% |
| TLE | TS Binary | 20 tkn | 25.6△13% | 14.7▼42% | 16.5▽27% | 16.3▼01% | 25.0△11% | 17.1▼32% | 22.1▽02% | 11.1▼50% | −31% |
| Best combination | | | 63.0△179% | 42.2▼33% | 67.6△199% | 20.0▼70% | 60.0△165% | 47.5▼25% | 60.9△169% | 56.4▼07% | −42% |
| Len | GB | sent. | 28.1△24% | 11.8▼58% | 21.1▽07% | 15.5▼26% | 30.4△34% | 18.4▼39% | 32.3△43% | 15.8▼51% | −44% |
| Len | LR | sent. | 19.6▽14% | 10.8▼45% | 17.0▽25% | 12.9▼24% | 22.2▽02% | 09.1▼59% | 22.9△01% | 09.1▼60% | −47% |
| Majority | — | | 15.4▽32% | | 13.0▽43% | | 15.9▽30% | | 17.7▽22% | | — |
| Approx. human | | global | 22.62 | | 22.62 | | 22.62 | | 22.62 | | — |

than GPT-2 (1.5B) on *RoFT*. For *RoFT-chatgpt*, the best perplexity estimator is a relatively small Phi-1.5 model, outperforming a GPT-2 estimator of comparable size by 15%. This impressive difference might be related to training data: Phi-1.5 was trained on GPT-3.5 generations. This suggests that small language models trained on data generated by LLMs, may appear a good perplexity-based detector of text generated by those particular LLMs. The baseline length-based classifier also improves its accuracy significantly (by 1.8x) when transferring to *RoFT-chatgpt*. We hypothesize that this kind of shallow feature emerges in ChatGPT generation and makes the task easier (see also Section 5). The other perplexity-based approach, DetectGPT, shows lower accuracy on both datasets. Mitchell et al. (2023) note that DetectGPT can detect whether the sample is generated by a specific base model, but we use several models in our setup, and text samples may be too short for this approach. On the other hand, DetectGPT has quite good MSE values, close to the regression approach that optimizes MSE directly.

**Cross-domain generalization**. Supervised ATD methods with fine-tuning such as RoBERTa are more sensitive to spurious correlations in a dataset and often demonstrate poor cross-domain transfer, especially compared to topology-based approaches (Tulchinskii et al., 2023a). Table 2 reports the results of cross-domain transfer between four text topics in the *RoFT-chatgpt* dataset. We report in-domain (IN) accuracy on domains seen during training and out-of-domain (OUT) accuracy on the unseen domain corresponding to this column; MSE scores are given in Table 5 (Appendix C). Each model was trained on three domains and tested on the fourth, unseen domain; we used 60% of these subsets mixed together as the training set, 20% as the validation set, and 20% as the test set for in-domain evaluation. Table 2 shows that RoBERTa's performance drops for all subsets very significantly, while perplexity-based classifiers demonstrate excellent cross-domain generalization for *Presidential Speeches* and *Short Stories*; for the *Recipes* domain, TTS classifiers prove to be the most stable, and the *New York Times* subset yields a large generalization gap for all classifiers. This means that every type of classifier can handle its own set of spurious features well, and no classifier is universally better than the others. Aggregation of different features can improve the results; for example, a classifier that returns the maximum of RoBERTa and perplexity-based predictions yields OOD scores on average 3% higher than the best result in Table 2 with comparable ID performance. As for classification accuracy, surprisingly, the perplexity-based classifier outperforms a fully fine-tuned RoBERTa on all subsets. Moreover, for *Recipes* the multilabel topological time series method places second; this happens because TDA-based methods are extremely stable under domain shift (despite being worse than others in absolute values on this dataset); see Section 5 for further discussion. In general, we conclude that perplexity-based classifiers outperform RoBERTa-based in cross-domain settings by a large margin. This is remarkable since the former is a simple classifier trained

Table 3: Cross-model transfer, original RoFT. Models are trained on all parts except one and tested in-domain (ID) on the same parts and out-of-domain (OOD) on the remaining part.

| Model | GPT2-XL | | GPT2 | | davinci | | ctrl-Politics | | ctrl-nocode | | tuned | |
|---|---|---|---|---|---|---|---|---|---|---|---|---|
| | ID | OOD | ID | OOD | ID | OOD | ID | OOD | ID | OOD | ID | OOD |
| RoBERTa + SEP | **46.4** | **40.9** | **46.0** | 08.8 | **63.3** | **19.7** | **49.4** | **59.1** | **50.2** | **60.6** | **49.6** | **23.9** |
| RoBERTa | 46.7 | 32.5 | 40.3 | 07.0 | 57.2 | 14.5 | 47.5 | 44.0 | 46.1 | 54.6 | 46.0 | 20.1 |
| *Phi-1.5 Perpl.* + GB | 26.0 | 27.0 | 31.5 | 12.2 | 30.5 | 15.3 | 30.8 | 23.9 | 30.3 | 19.2 | 34.7 | 12.9 |
| *Phi-2 Perpl.* + GB | 34.1 | 26.4 | 32.0 | 17.6 | 35.9 | 16.2 | 34.1 | 19.5 | 34.3 | 23.2 | 37.2 | 12.4 |
| PHD + TS ML | 31.8 | 04.0 | 25.6 | 04.5 | 31.4 | 02.1 | 25.7 | 08.2 | 25.2 | 11.1 | 21.4 | 12.6 |
| TLE + TS Binary | 14.2 | 03.2 | 12.4 | 07.8 | 14.5 | 01.0 | 11.0 | 05.0 | 11.5 | 04.0 | 14.5 | 07.0 |
| *Length* + GB | 21.7 | 04.8 | 18.3 | 01.2 | 19.2 | 06.3 | 17.3 | 03.8 | 18.1 | 00.0 | 21.6 | 00.0 |
| Human Baseline | 22.5 | 17.2 | 22.6 | **22.5** | 24.7 | 14.1 | 22.6 | 21.6 | 22.6 | 23.9 | 21.5 | 25.9 |

on ten features extracted by a LM with frozen weights, while the latter involves full LM fine-tuning. As for the length-based baseline, for in-domain data average sentence length provides a strong signal, leading to accuracy of 20–32% and even outperforming topological methods. But cross-domain generalization fails, which means that we should prefer classifiers that ignore this feature to achieve good generalization (see also Section 5).

**Cross-model generalization**. We tune our classifiers on generation results produced by one model and test the performance for all other models. In general, this task is harder for all considered classifiers: there are models for which prediction accuracy drops down to virtually zero values (detailed experimental results are shown in Appendix D: Tables 3, 6, 7). But we observe an interesting result for the perplexity-based classifier: it achieves good generalization when transferring to very large models such as *GPT3-davinci* and *GPT2-XL*, while for other models the generalization is poor. In general, the strongest classifiers fail to generalize to *simpler* models; e.g., RoBERTa-based classifiers underperform on the generalization to GPT-2 and baseline; the baseline is also the hardest subset for perplexity-based classifiers (see Section 5 for a discussion).

## 5 Discussion and analysis of the results

**Detection models**. We have found that although RoBERTa-based classifiers demonstrate excellent results for in-domain classification, they lose to perplexity-based methods when tested on texts with new styles and topics not present in the training set. For cross-model generalization, all methods demonstrate insufficient generalization abilities, especially when tested on generations of much larger or much smaller models. This suggests that AI content detectors can be fooled by either too good or *too bad* artificial texts. For perplexity-based classifiers, we have found that the size of the base model is not the main requirement for a good and robust detector. The training dataset for perplexity estimation is more important; e.g., ChatGPT generations were best detected by a much smaller model pretrained on synthetic data generated by the same model family. The above observations suggest that pre-training of the perplexity estimators on a large amount of synthetic data from different generators may become the best choice for AI-generated content detection. We leave it for future research. Topological classifiers demonstrated high robustness over domain transfer, but not very high overall accuracy. Geometric properties of the embeddings for both RoFT and *RoFT-chatgpt*, however, show the difference between PHD distributions of real and fake RoFT text for different models (Fig. 10) and topics (Figs. 11, 12 in Appendix E). TLE dimension distributions for sentences in human-written and AI-generated parts of the texts are different as well (Figs. 13, 14, 15, Appendix E). This means that topological classifiers may help in robust artificial content detection in some particular domains and models.

**Data properties**. Next, we summarize the observed properties of the data that influence detection quality and create difficulties for different types of artificial content detectors.

1. First, the *length of sentences* seems to play an important role, deceiving our classifiers. There is indeed a significant difference between distributions of sentence lengths written by humans and generated by LLMs (see Fig. 2 and Figs. 5, 6 in Appendix E). This is supported

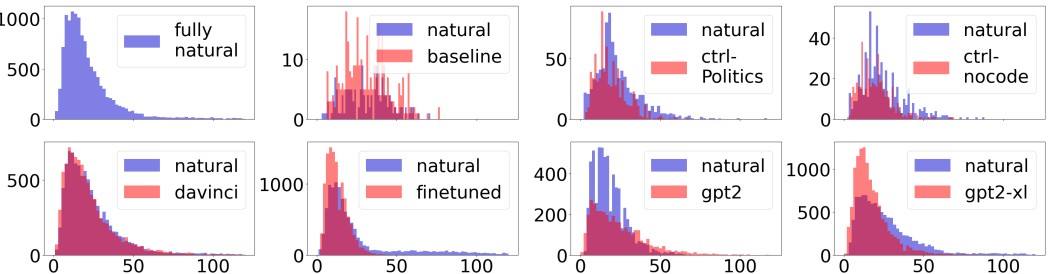

Figure 2: Sentence length distributions in RoBERTA tokens, original RoFT, by model

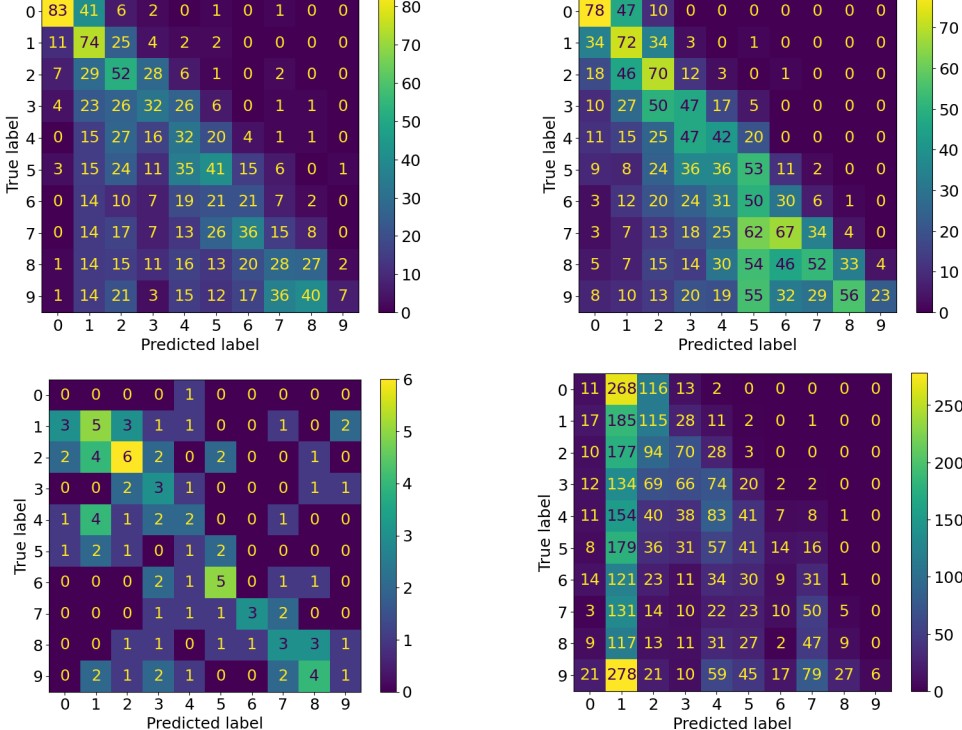

Figure 3: Confusion matrices of RoBERTa predictions on the four domains in the *RoFT-chatgpt* dataset: New York Times, Short Stories, Presidental Speeches, Recipes (left to right).

by our experiments with the length-based classifier, which sometimes outperforms other methods in terms of accuracy (Tables 1 and 2), but fails to generalize.

2. *Label distributions* (see Figs. 8 and 9 in Appendix E) vary significantly across models. GPT-2 is especially different from natural texts, and the behavior of our models in cross-model transfer to GPT-2 (Appendix D), suggests that the only model stable under the label distribution shift is perplexity-based regression. Although its accuracy numbers are low, it outperforms the human level in terms of MSE even in out-of-domain evaluation.

3. Special *text structure* can hurt ATD: e.g., recipes often contain numbered lists, and the first sentence tends to be comparatively long, and as a result the *Recipes* topic is hard for all classifiers to generalize to. The RoBERTa classifier loses the most here, so we investigated its confusion matrices; Fig. 3 shows that RoBERTa tends to classify all examples in the *Recipes* topic as fully generated. This kind of text structure can be viewed as an *adversarial example* for AI-generated texts, and indeed, ChatGPT is known for its tendency to generate structured output with bullet lists.

4. *Semantic and grammar inconsistencies.* Surprisingly, cross-model generalization of RoBERTa-based and perplexity-based classifiers fails not only for the *best* OOD genera-

tor (davinci), but also for the *worst* generators: small-size GPT-2 and baseline. Perplexity distributions for texts generated by *baseline* or GPT-2 are very different from the distributions of texts generated by other models (see Fig. 16). This might explain the poor performance of decision trees and linear classifiers when these generators are excluded from the training set. On the other hand, both generators are among the easiest for human raters, because of the high level of semantic inconsistencies and grammar issues. It seems that the supervised classifier discovers non-regularities in the text as a feature of human-generated texts; indeed, they present in our training set as a part of first-person narratives, especially in the Short Stories dataset.

5. *Discourse structure*. Finally, analysing the poor generalization to narrative domains, we discover a subtle property of human-generated texts that influences boundary detection, namely the underlying discourse structure of the story that can be measured as peaks of perplexity along the story (Fig. 17, Appendix E). Human-generated texts tend to have 2-3 such peaks, e.g. a peak in the middle of the story corresponding to a plot twist or change of narrative focus. This leads to the difficulties in perplexity-based boundary detection: the perplexity drop in the beginning of the fake text part is confused with the same drop due to the story discourse change. Strong perplexity-based classifiers can handle this issue due to the *std* feature reflecting token-level perplexity deviation (Fig. 18, Appendix E). Fig. 2b shows that the RoBERTa-based classifier is prone to this discourse-related issue: it often incorrectly predicts the real-fake boundary in the middle of a human-generated story (5-6th sentence).

A full analysis of feature distributions across domains and models is given in the Appendix.

# 6 Conclusion

In this work, we address the task of boundary detection between human-written and generated parts in texts that combine both. We believe that this setting is increasingly important in real world applications and is a natural setting for recognizing artificial text since it is often mixed with and prompted by human text. We have considered the RoFT dataset, presented its modification *RoFT-chatgpt* generated with a more modern LLM, and investigated the performance of features that were useful for artificial text detection in previous works. In particular, we have shown that LLM fine-tuning works reasonably well for this task but tends to overfit to spurious features in the data, which leads to generalization failures in some settings. On the other hand, perplexity-based and topological features provide a signal that can help in these situations. We have demonstrated that perplexity features are the best overall on balance between accuracy, generalization, and training complexity.

We analyze the base models for perplexity estimation and conclude, that reasonably small models work well, but pre-training data is important. Including synthetic data in the pre-training dataset improves the performance of the detector. As for the limitations of our approach, boundary detection is very hard in cross-domain and cross-model settings, both for short texts such as RoFT (due to lack of information) and longer texts (due to a large space of possibilities). Therefore, it is no wonder that none of the methods have achieved a really high quality in this setting, and the results suggest a large room for improvement. Besides, all methods we considered were based on Transformers with relatively small context window size, which limits the transferability of the proposed approaches onto longer text samples. Finally, our analysis has uncovered gaps in current approaches and discovered the more difficult aspects of the task, which we plan to address in future research.

**Acknowledgments**

The work of Sergey Nikolenko was performed at the Saint Petersburg Leonhard Euler International Mathematical Institute and supported by the Ministry of Science and Higher Education of the Russian Federation (agreement no. 075-15-2022-289 dated 06/04/2022).

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

## A  Intrinsic Dimension Estimation Methods

According to the manifold hypothesis (Narayanan & Mitter, 2010), the data $X$ lies on a low-dimensional submanifold: $X \subseteq M^n \subseteq R^d$, where $d$ is the extrinsic dimension and $n$ is the intrinsic dimension (ID). The geometric and topological properties of the manifold $M$ are of particular interest. There are various methods for estimating the ID that can be divided into global and local methods.

For the tight local intrinsic dimension estimator (TLE) proposed by Amsaleg et al. (2019), we use the neighborhood center point $x$, a set of neighborhood samples $V$ and a specially defined distance between points in a sufficiently small neighborhood of $x$:

$$d_{x,r}(q,v) = \frac{r(v-q) \cdot (v-q)}{2(x-q) \cdot (v-q)},$$

$$(1)$$

where $r$ is the radius of the neighbourhood. For every three points $x, v, w$ we can compute

$$M(x,v,w) = \ln \frac{d_{x,r}(v,w)}{r} + \ln \frac{d_{x,r}(2x-v,w)}{r}.$$

If $V_* = V \cup \{x\}$, then the intrinsic dimension can be found by averaging the estimates for all points $x$, as defined by the following formula:

$$\hat{m}_r(x) = -\left( \frac{1}{|V_*|^2} \sum_{v,w \in V_*,\ v \neq w} M(x,v,w) \right)^{-1}$$

$$(2)$$

Applied algebraic topology provides effective tools for analyzing the topological structure of data. The theoretical foundations of topological data analysis (TDA) have been described in detail by, e.g., Barannikov (1994) and Carlsson (2020). TDA allows us to consider a dataset $X \subseteq R^d$ from the topological point of view. In order to move from point clouds $X$ to

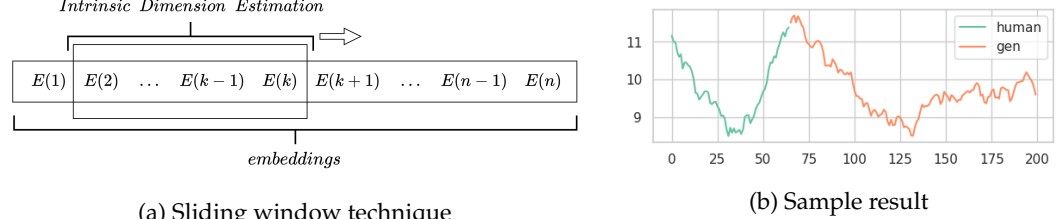

(a) Sliding window technique

(b) Sample result

Figure 4: Multilabel time series classification: (a) estimating the intrinsic dimension of token embeddings in a sliding window, (b) a sample resulting series: green – human-written, orange – machine-generated tokens.

topological spaces, it is necessary to approximate the data by a simplicial complex $R$. In our research, we use the Vietoris-Rips complex $R(X;t)$. The method of constructing the complex $R$ is as follows: simplexes are formed by subsets of points from X whose pairwise distances do not exceed $t$ (a scaling parameter). An increasing sequence of simplicial complexes is called a filtration: $\{R_t\}_{(t\geq 0)} = R_{t_0} \subseteq R_{t_1} \subseteq ... \subseteq R_s$.

Homology groups $H_i(R)$ are a topological invariant that expresses the properties of a topological space $R$. We use $\beta_i(R) = \dim H_i(R)$, which is known as the $i$th *Betti number*, a topological feature equal to the dimension of the homology group; for $i = 0, 1, 2$ the Betti number corresponds to the number of connectivity components, cycles, and cavities respectively.

Topological features appear and disappear at different values of $t$, which leads to the next core concept in TDA: the *barcode*. It summarises the dynamics of topological features in the filtration process. A *bar* is the lifetime of the $n$th homology feature $I_n = t_n^{\text{birth}} - t_n^{\text{death}}$. A long bar in the barcode means that the data contains a fairly persistent and informative topological feature.

Schweinhart (2020) introduced the persistent homological fractal dimension (PHD) that generalized Steele (1988) for higher dimensions of the homology group and used the topological properties of the point cloud. The PHD has already been proven to be useful in the study of the properties of deep learning models (Birdal et al., 2021; Magai, 2023).

Let us denote the power-weighted sum of $N$ bars for the $i$th degree of homology as follows:

$$E_\alpha^i(X) = \sum_{i=1}^{N} I_i^\alpha. \tag{3}$$

It is interesting to note that $E_1^0(X_n)$ is equal to the length of the Euclidean minimum spanning tree (MST) of $X_n \subseteq R^d$ (Skraba et al., 2017).

Then the persistent homological fractal dimension (PHD) can be defined as follows:

$$\text{PHD}^i = \frac{\alpha}{1 - \beta}, \tag{4}$$

where

$$\beta = \lim_{n\to\infty} \sup \frac{\log(\mathbb{E}(E_\alpha^i(x_1,...,x_n)))}{\log(n)}, \tag{5}$$

and $x_1, \ldots, x_n$ are sampled independently from $X$. That is, $\text{PHD}^i(X) = d$ if $E_\alpha^i(x_1,...,x_n)$ scales as $n^{\frac{d-\alpha}{d}}$ and $\alpha \geq 0$ (we take $i = 0, \alpha = 1$). Persistent homological fractal dimension can be estimated by analyzing the asymptotical behavior at $n \to \infty$ of $E_\alpha^i(x_1...x_n)$ for every $i$. In other words, to calculate PHD we must find a power law that shows how $E_\alpha^i(x_1...x_n)$ scales as $n$ increases. See Adams et al. (2020) and Schweinhart (2020) for more details.

Table 4: Accuracy for leave-one-out cross-domain evaluation on *RoFT-chatgpt*, full table.

| Pre-dictor | Model | Context | Presidential Speeches IN ↑ | OUT ↑ | Recipes IN ↑ | OUT ↑ | New York Times IN ↑ | OUT ↑ | Short Stories IN ↑ | OUT ↑ |
|---|---|---|---|---|---|---|---|---|---|---|
| Text | RoBERTa SEP | global | **57.3** | 31.4 | 43.2 | 13.1 | 53.2 | 38.1 | **54.3** | 28.6 |
| Text | RoBERTa | global | 54.2 | 40.2 | 39.7 | 15.1 | **53.3** | 34.0 | 50.3 | 27.7 |
| Perpl. | GPT2, GB | sentence | 36.1 | 35.3 | 36.6 | 19.4 | 36.9 | 29.7 | 32.8 | 32.9 |
| Perpl. | GPT2, LR | sentence | 34.8 | 32.4 | 40.6 | 20.4 | 36.7 | 28.9 | 32.7 | 35.6 |
| Perpl. | GPT2, Regr. (GB) | sentence | 20.9 | 22.6 | 23.8 | 14.4 | 18.5 | 16.1 | 21.2 | 21.8 |
| Perpl. | Phi1, GB | sentence | 36.2 | 24.5 | 26.5 | 21.6 | 39.1 | 23.8 | 42.1 | 21.2 |
| Perpl. | Phi1, LR | sentence | 33.6 | 27.4 | 25.9 | 26.2 | 37.4 | 25.5 | 35.5 | 22.9 |
| Perpl. | **Phi1.5, GB** | sentence | 52.5 | **52.0** | 60.4 | 24.1 | 52.5 | **45.7** | 48.7 | **56.1** |
| Perpl. | Phi1.5, LR | sentence | 48.7 | 41.2 | 60.9 | 21.1 | 50.8 | 45.2 | 47.1 | 51.5 |
| Perpl. | Phi2, GB | sentence | 50.1 | 46.1 | 55.6 | 22.8 | 55.0 | 42.1 | 48.1 | 53.9 |
| Perpl. | Phi2, LR | sentence | 49.6 | 42.1 | 56.4 | **24.4** | 50.5 | 43.4 | 48.5 | 52.0 |
| Perpl. | LLaMA2-7B, GB | sentence | 41.4 | 25.5 | 39.0 | 17.7 | 45.2 | 24.5 | 41.1 | 27.4 |
| Perpl. | LLaMA2-7B, LR | sentence | 34.5 | 15.7 | 37.9 | 14.8 | 37.2 | 23.5 | 36.1 | 24.4 |
| PHD | TS multilabel | 100 tokens | 20.3 | 13.7 | 19.2 | 19.5 | 20.9 | 17.2 | 21.2 | 17.6 |
| TLE | TS Binary | 20 tokens | 25.6 | 14.7 | 16.5 | 16.3 | 25.0 | 17.1 | 22.1 | 11.1 |
| Length | GB | sentence | 28.1 | 11.8 | 21.1 | 15.5 | 30.4 | 18.4 | 32.3 | 15.8 |
| Length | LogRegr | sentence | 19.6 | 10.8 | 17.0 | 12.9 | 22.2 | 09.1 | 22.9 | 09.1 |
| Majority | | — | 15.4 | | 13.0 | | 15.9 | | 17.7 | |
| Approximated human | | global | 22.62 | | 22.62 | | 22.62 | | 22.62 | |

## B  gpt-3.5-turbo and gpt-4 zero-shot experiment details

Below we show the prompt used for gpt-3.5-turbo and gpt-4 zero-shot experiments.

You will be provided with a text that contains human–written sentences. Your task is to analyze the text and determine if any of the sentences were generated by an AI system.
You should output a single number between 0 and 9 to indicate where the first AI–generated sentence occurs in the text (if any):
For your output, you should provide a single number between 0 and 9, with the following meaning:
0 – The entire text was generated by an AI system.
1 to 8 – The first AI–generated sentence occurs at that position in the text (1 for the first sentence, 2 for the second sentence, and so on).
9 – The entire text was written by a human with no AI–generated sentences.
Your output should consist solely of this single digit number, with no additional text or explanation:
{content}

## C  Cross-domain transfer

Tables 5 and 4 supplement Table 2 from the main text, reporting the results of cross-domain transfer for our methods on the *RoFT-chatgpt* dataset across four text topics present in the data. We report in-domain and out-of-domain accuracy: the IN column shows results from domains seen during training, while the OUT column reflects the model's ability to detect artificial texts in the unseen domain corresponding to this column. For each model, training was done on three out of the four domains, and the resulting model was tested on the fourth, unseen domain; we used 60% of these subsets, mixed together, as the training set, 20% as the validation set, and 20% as the test set for in-domain evaluation.

Table 5: Mean squared errors from leave-one-out cross-domain evaluation on *RoFT-chatgpt*, full table.

| Pre-dictor | Model | Context | Presidential Speeches IN ↓ | OUT ↓ | Recipes IN ↓ | OUT ↓ | New York Times IN ↓ | OUT ↓ | Short Stories IN ↓ | OUT ↓ |
|---|---|---|---|---|---|---|---|---|---|---|
| Text | RoBERTa SEP | global | 02.6 | 10.6 | 02.6 | 18.3 | 03.4 | 07.9 | 02.3 | 09.0 |
| Text | RoBERTa | global | 02.3 | 07.5 | 02.8 | 13.5 | 02.9 | 06.2 | 02.6 | 05.5 |
| Perpl. | GPT2, GB | sentence | 07.3 | 08.9 | 07.0 | 14.6 | 07.2 | 09.4 | 08.5 | 08.5 |
| Perpl. | GPT2, LR | sentence | 08.2 | 11.5 | 06.0 | 16.8 | 08.7 | 11.8 | 09.6 | 09.3 |
| Perpl. | GPT2, Regr. (GB) | sentence | 04.7 | 06.6 | 04.8 | 09.7 | 04.9 | 05.7 | 05.2 | 05.0 |
| Perpl. | Phi1, GB | sentence | 09.3 | 11.8 | 10.8 | 15.1 | 08.5 | 14.2 | 08.7 | 14.7 |
| Perpl. | Phi1, LR | sentence | 09.8 | 11.9 | 11.5 | 12.9 | 08.9 | 12.2 | 09.1 | 13.5 |
| Perpl. | Phi1.5, GB | sentence | 04.5 | 08.5 | 03.0 | 11.8 | 03.9 | 06.3 | 05.6 | 03.4 |
| Perpl. | Phi1.5, LR | sentence | 04.5 | 09.3 | 03.6 | 12.2 | 04.0 | 07.2 | 05.2 | 03.0 |
| Perpl. | Phi2, GB | sentence | 04.3 | 07.3 | 03.0 | 15.5 | 04.3 | 06.6 | 04.6 | 04.3 |
| Perpl. | Phi2, LR | sentence | 04.5 | 09.9 | 03.4 | 13.6 | 04.3 | 07.5 | 05.4 | 03.0 |
| Perpl. | Phi2, GB | sentence | 04.3 | 07.3 | 03.0 | 15.5 | 04.3 | 06.6 | 04.6 | 04.3 |
| Perpl. | Phi2, LR | sentence | 04.5 | 09.9 | 03.4 | 13.6 | 04.3 | 07.5 | 05.5 | 03.0 |
| Perpl. | LLAMA-2-7B, GB | sentence | 05.0 | 14.0 | 05.5 | 13.2 | 05.3 | 11.4 | 07.3 | 09.7 |
| Perpl. | LLAMA-2-7B, LR | sentence | 06.3 | 18.3 | 06.0 | 19.5 | 06.1 | 08.8 | 08.3 | 05.0 |
| PHD | TS multilabel | 100 tokens | 12.3 | 14.6 | 10.7 | 11.0 | 14.1 | 16.8 | 11.6 | 11.6 |
| TLE | TS Binary | 20 tokens | 12.3 | 15.6 | 18.0 | 15.4 | 12.0 | 17.6 | 17.4 | 23.9 |
| Length | GB | sentence | 12.8 | 15.9 | 14.2 | 17.9 | 13.7 | 18.4 | 12.5 | 15.3 |
| Length | LogReg | sentence | 20.1 | 20.5 | 16.7 | 24.7 | 17.2 | 22.1 | 18.5 | 22.9 |
| Majority | | — | 27.5 | | 27.4 | | 27.9 | | 28.0 | |
| Approximated human** | | global | 13.88 | | 13.88 | | 13.88 | | 13.88 | |

## D   Cross-model transfer

Tables 6 and 7 show our experimental results on cross-model transfer for all considered text generation models. The artificial text boundary detection models were trained on the parts of the dataset generated by all language models except one, which is held out for cross-model testing, and tested in-domain (ID) on the same parts and out-of-domain (OOD) on the remaining part generated by the held-out model.

## E   Additional Dataset Analysis

In this section we provide additional statistics and visualizations for the distributions of various features in the data. In particular, we note that on most diagrams, real texts have smaller PHD than fake texts, which is a very different result from the statistics presented by Tulchinskii et al. (2023a), who noted that the PHD of real texts is larger than that of fake texts. We hypothesize that it can be due either to very short lengths of texts in our work compared to the texts considered by Tulchinskii et al. (2023a) or due to differences in the sampling strategy used by Dugan et al. (2020) and Tulchinskii et al. (2023a) when generating texts. Another observation is that the TLE dimension is very different for all generator models in the original RoFT dataset. This may be the reason for the bad generalization performance of intrinsic dimension-based algorithms across domains. For *RoFT-chatgpt* PHD and TLE, real and fake texts are close to each other.

We show dataset statistics in the following figures:

- Figures 2, 5, and 6 show the lengths of texts in tokens produced by the standard RoBERTa tokenizer (the figures have a cutoff of 100 for readability but the datasets do contain a few longer sentences);

Table 6: Original RoFT, cross-model transfer, part 1. The models were trained on all parts of the dataset except one and tested in-domain (ID) on the same parts and out-of-domain (OOD) on the remaining part.

| Model | Metric | GPT2-XL | | GPT2 | | davinci | |
|---|---|---|---|---|---|---|---|
| | | ID | OOD | ID | OOD | ID | OOD |
| RoBERTa + SEP | Acc, % | 46.38 | 40.94 | 45.95 | 08.78 | 63.25 | 19.73 |
| RoBERTa + SEP | SoftAcc1, % | 76.85 | 76.83 | 76.71 | 31.22 | 85.52 | 47.27 |
| RoBERTa + SEP | MSE | 03.90 | 02.92 | 04.17 | 06.77 | 03.04 | 07.59 |
| RoBERTa | Acc, % | 46.68 | 32.56 | 40.30 | 06.94 | 57.20 | 14.48 |
| RoBERTa | SoftAcc1, % | 77.30 | 72.07 | 75.52 | 25.31 | 84.61 | 40.49 |
| RoBERTa | MSE | 03.73 | 03.10 | 03.70 | 07.18 | 02.94 | 08.56 |
| *GPT-2 Perplexity* + GB | Acc, % | 23.00 | 23.43 | 28.12 | 04.08 | 23.75 | 19.78 |
| *GPT-2 Perplexity* + GB | SoftAcc1, % | 40.35 | 47.90 | 47.96 | 30.61 | 46.03 | 42.46 |
| *GPT-2 Perplexity* + GB | MSE | 15.39 | 10.51 | 12.19 | 15.99 | 11.85 | 15.38 |
| *GPT-2 Perplexity* + LogReg | Acc, % | 21.27 | 08.86 | 23.67 | 03.47 | 21.43 | 08.31 |
| *GPT-2 Perplexity* + LogReg | SoftAcc1, % | 33.33 | 22.14 | 39.80 | 27.45 | 35.35 | 25.19 |
| *GPT-2 Perplexity* + LogReg | MSE | 21.52 | 24.48 | 16.99 | 18.98 | 19.71 | 22.06 |
| *GPT-2 Perplexity* + Regr | Acc, % | 11.68 | 15.78 | 14.56 | 14.18 | 13.91 | 15.30 |
| *GPT-2 Perplexity* + Regr | SoftAcc1, % | 34.84 | 49.37 | 39.36 | 47.86 | 46.10 | 44.43 |
| *GPT-2 Perplexity* + Regr | MSE | 07.67 | 04.62 | 06.80 | 06.40 | 06.73 | 06.64 |
| PHD + TS ML | Acc, % | 31.84 | 04.02 | 25.64 | 04.49 | 31.38 | 02.05 |
| PHD + TS ML | SoftAcc1, % | 56.08 | 14.14 | 44.28 | 35.85 | 53.31 | 13.82 |
| PHD + TS ML | MSE | 11.13 | 28.18 | 16.37 | 10.74 | 11.82 | 27.03 |
| *TLE* + TS Binary | Acc, % | 14.17 | 03.15 | 12.36 | 07.76 | 14.48 | 00.98 |
| *TLE* + TS Binary | SoftAcc1, % | 31.14 | 13.10 | 29.19 | 32.14 | 34.01 | 11.58 |
| *TLE* + TS Binary | MSE | 21.58 | 28.20 | 21.36 | 16.35 | 18.12 | 30.45 |
| *Length* + GB | Acc, % | 21.70 | 04.75 | 18.30 | 01.22 | 19.18 | 06.33 |
| *Length* + GB | SoftAcc1, % | 36.00 | 09.48 | 33.20 | 22.02 | 36.09 | 22.07 |
| *Length* + GB | MSE | 23.56 | 33.32 | 27.00 | 19.27 | 17.35 | 20.00 |
| Human Baseline | Acc, % | 22.48 | 17.23 | 22.59 | 22.53 | 24.74 | 14.06 |
| Human Baseline | SoftAcc1, % | 41.91 | 37.03 | 39.73 | 48.01 | 42.44 | 33.47 |
| Human Baseline | MSE | 13.49 | 14.69 | 14.29 | 09.86 | 14.03 | 12.91 |

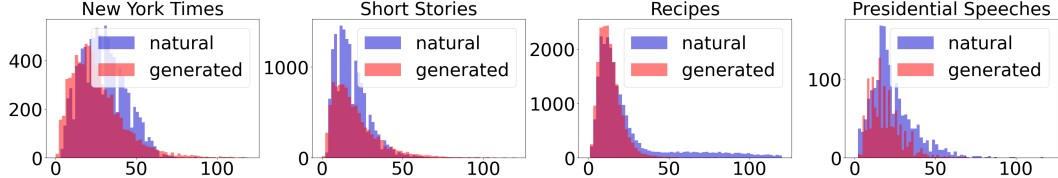

Figure 5: Sentence length distributions in RoBERTA tokens, original RoFT, by topic

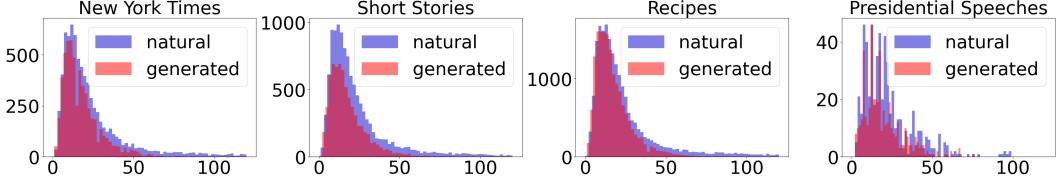

Figure 6: Sentence length distributions in RoBERTA tokens, RoFT-chatgpt, by topic

- Figure 7 shows the distribution of pretrained (but not fine-tuned) RoBERTa [CLS] embeddings for real and fake parts of text samples from the original RoFT and *RoFT-chatgpt* datasets;

Table 7: Original RoFT, cross-model transfer, part 2. The models were trained on all parts of the dataset except one and tested in-domain (ID) on the same parts and out-of-domain (OOD) on the remaining part.

| Model | Metric | ctrl-Politics | | ctrl-nocode | | tuned | | baseline | |
|---|---|---|---|---|---|---|---|---|---|
| | | ID | OOD | ID | OOD | ID | OOD | ID | OOD |
| RoBERTa + SEP | Acc, % | 49.35 | 59.12 | 50.20 | 60.61 | 49.55 | 23.85 | 51.35 | 06.35 |
| RoBERTa + SEP | SoftAcc1, % | 78.49 | 89.31 | 80.10 | 84.85 | 81.55 | 56.96 | 79.84 | 15.87 |
| RoBERTa + SEP | MSE | 02.86 | 01.24 | 02.93 | 01.87 | 02.33 | 06.33 | 02.51 | 39.32 |
| RoBERTa | Acc, % | 47.46 | 44.03 | 46.07 | 54.55 | 45.91 | 20.98 | 47.29 | 04.76 |
| RoBERTa | SoftAcc1, % | 78.43 | 85.53 | 78.01 | 86.87 | 80.13 | 52.88 | 78.49 | 15.87 |
| RoBERTa | MSE | 02.80 | 01.21 | 02.79 | 01.07 | 02.37 | 06.49 | 02.69 | 36.00 |
| *GPT-2 Perpl.* + GB | Acc, % | 25.27 | 10.69 | 25.32 | 07.07 | 30.88 | 10.15 | 24.44 | 06.35 |
| *GPT-2 Perpl.* + GB | SoftAcc1, % | 48.89 | 27.04 | 47.71 | 24.24 | 53.87 | 29.20 | 47.64 | 14.29 |
| *GPT-2 Perpl.* + GB | MSE | 11.81 | 20.96 | 12.27 | 22.70 | 11.67 | 17.86 | 12.01 | 39.62 |
| *GPT-2 Perpl.* + LogReg | Acc, % | 24.08 | 07.55 | 21.88 | 07.07 | 28.50 | 08.04 | 24.77 | 03.17 |
| *GPT-2 Perpl.* + LogReg | SoftAcc1, % | 42.23 | 23.90 | 38.84 | 19.19 | 42.78 | 22.28 | 42.45 | 15.87 |
| *GPT-2 Perpl.* + LogReg | MSE | 15.70 | 23.27 | 16.69 | 25.38 | 17.46 | 24.20 | 15.72 | 40.86 |
| *GPT-2 Perpl.* + Regr | Acc, % | 14.80 | 13.21 | 13.96 | 14.14 | 15.70 | 11.90 | 13.62 | 03.17 |
| *GPT-2 Perpl.* + Regr | SoftAcc1, % | 42.06 | 40.25 | 41.89 | 33.33 | 43.08 | 36.03 | 39.25 | 14.29 |
| *GPT-2 Perpl.* + Regr | MSE | 06.40 | 07.78 | 06.55 | 08.31 | 06.81 | 07.10 | 06.84 | 22.86 |
| PHD + TS ML | Acc, % | 25.70 | 08.18 | 25.18 | 11.11 | 21.44 | 12.58 | 23.13 | 03.70 |
| PHD + TS ML | SoftAcc1, % | 47.33 | 32.70 | 47.30 | 39.39 | 35.31 | 26.37 | 45.64 | 05.56 |
| PHD + TS ML | MSE | 14.09 | 11.23 | 13.62 | 09.41 | 18.28 | 18.43 | 13.42 | 52.39 |
| *TLE* + TS binary | Acc, % | 10.98 | 05.03 | 11.53 | 04.04 | 14.51 | 06.56 | 12.39 | 03.18 |
| *TLE* + TS binary | SoftAcc1, % | 29.08 | 18.87 | 28.21 | 22.22 | 30.21 | 17.26 | 28.21 | 15.88 |
| *TLE* + TS binary | MSE | 20.36 | 23.28 | 20.84 | 21.82 | 21.82 | 26.64 | 20.59 | 25.55 |
| *Length* + GB | Acc, % | 17.32 | 03.77 | 18.05 | 0.0 | 21.64 | 0.04 | 15.24 | 26.98 |
| *Length* + GB | SoftAcc1, % | 33.96 | 18.86 | 35.71 | 15.15 | 35.80 | 10.87 | 32.56 | 34.92 |
| *Length* + GB | MSE | 22.48 | 23.87 | 24.16 | 26.53 | 23.59 | 32.78 | 16.25 | 16.0 |
| Human Baseline | Acc, % | 22.60 | 21.6 | 22.57 | 23.94 | 21.46 | 25.90 | 22.41 | 46.15 |
| Human Baseline | SoftAcc1, % | 40.61 | 41.6 | 40.59 | 45.07 | 39.17 | 44.92 | 40.46 | 63.46 |
| Human Baseline | MSE | 13.87 | 10.57 | 13.87 | 07.70 | 13.92 | 13.48 | 13.82 | 11.51 |

- Figure 8 shows the distribution of labels in the original RoFT dataset by generator;
- Figure 9 shows the distribution of labels in the original RoFT dataset by topic; this distribution is identical to the corresponding distribution for the *RoFT-chatgpt* dataset;
- Figure 10 shows the distribution of PH dimensions of real and fake parts of the text by generator;
- Figures 11 and 12 show the distributions of PH dimensions by topic for the original RoFT and *RoFT-chatgpt* respectively;
- Figure 13 shows the distribution of TLE dimensions of different sentences by generator;
- Figures 14 and 15 show the the distributions of TLE dimensions by topic for original RoFT and *RoFT-chatgpt* respectively.

# F   Detailed experimental results

In this section, we provide additional statistics and visualizations regarding our experimental results. Figure 17 visualizes the changes in perplexities for sentences from the texts in *RoFT-chatgpt* by their labels. We make the following observations.

First, perplexities of the first couple of sentences across all texts are quite high, and the average perplexity of sentences decreases by the end of the text. This is probably due to

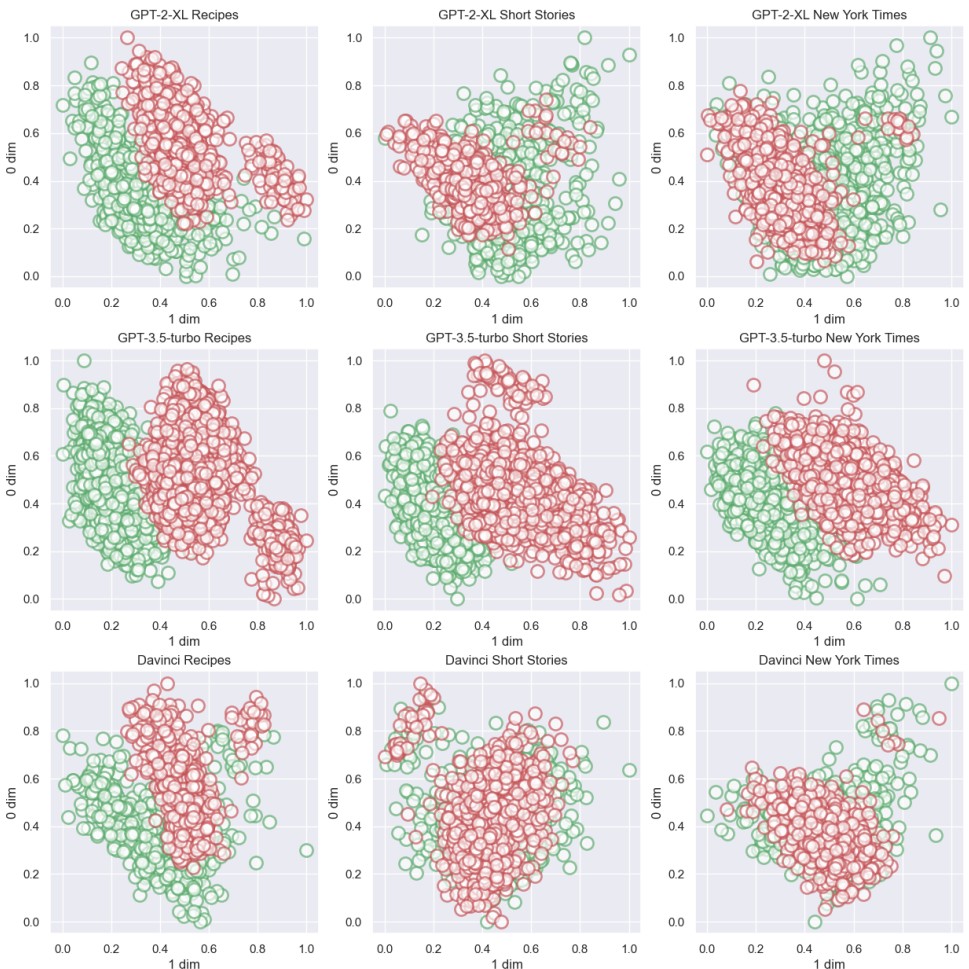

Figure 7: Distribution of pretrained (but not fine-tuned) RoBERTa [CLS] embeddings of real and fake parts of text samples from the original RoFT and *RoFT-chatgpt* datasets. The dimension is reduced to 2D via principal component analysis (F.R.S., 1901).

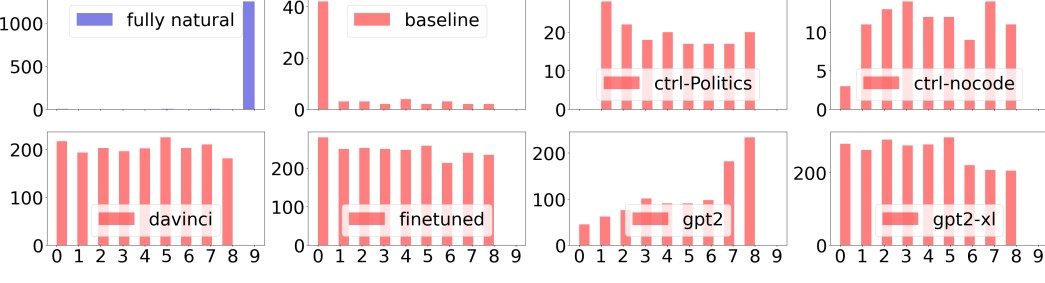

Figure 8: Label distributions for the original RoFT dataset by model

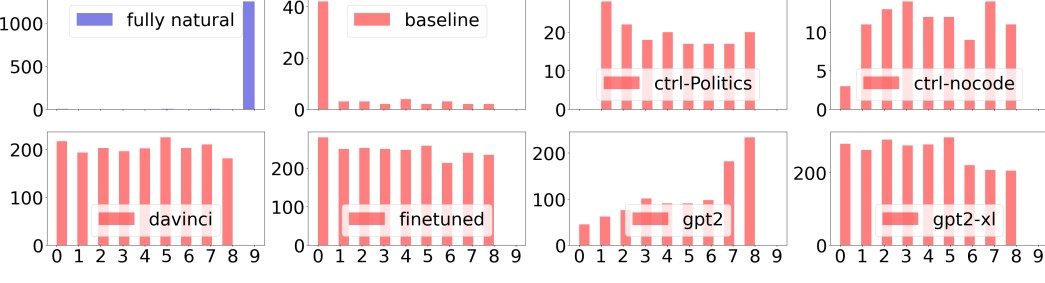

Figure 9: Label distributions for the original RoFT dataset by topic

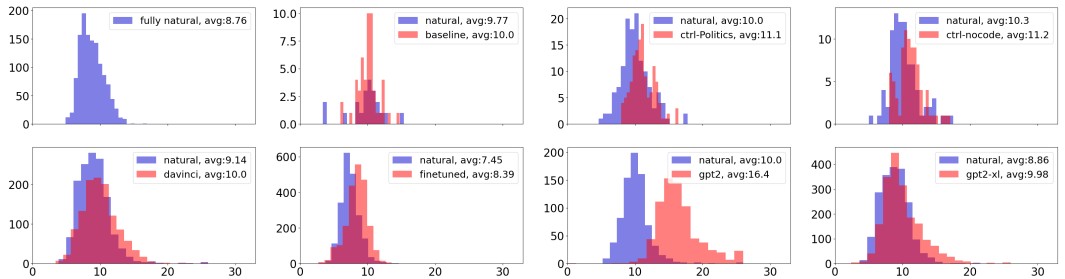

Figure 10: PHD distributions for the real and fake parts of the RoFT dataset, by generator models

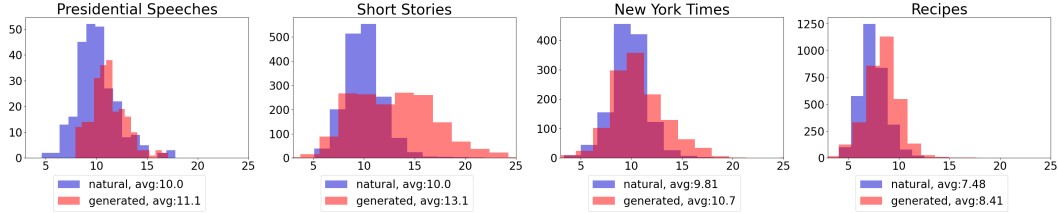

Figure 11: PHD distributions for the real and fake parts of the RoFT dataset, by topics

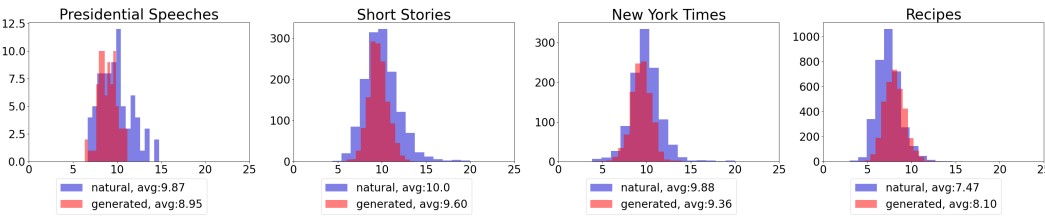

Figure 12: PHD distributions for the real and fake parts of the *RoFT-chatgpt* dataset text, by topics

the fact that for the words of the first sentences the length of the text prefix is not enough for a stable calculation of perplexity. One solution to mitigate this effect and hence make perplexity-based classifiers more stable might be to generate new prefixes for the text using some generative model (e.g. *gpt-3.5*) and calculate perplexities of original text words using this generated prefix. We leave this idea for further research. Figure 18 visualizes the coefficients of a logistic regression model trained on sentence perplexities from the *RoFT-chatgpt* dataset (*Perplexity + LogRegr* rows in the tables). We can see a distinct pattern in this figure. For the label $k$, which means that the first fake sentence in the text is the $(k + 1)$st, the highest value of the coefficient are $k$th and $k_{10}$th ones, and the lowest ones are often $(k + 2)$nd and $(k + 12)$th. This could mean that the model is "searching" for a sudden drop of mean and variance of perplexity at a point where the fake part is starting. This fits together well with the idea that language model (Phi-1.5) trained on data from another model (GPT-3.5 version) see text generated by this model as a more "natural" one than real human-produced text. Therefore, perplexity often drops at the point where fake text begins, and logistic regression can pick up this effect and use it as a decision rule.

Finally, Figure 18b visualizes the confusion matrix on the test set of a logistic regression trained on the RoFT-ChatGPT dataset. We see that most of the errors concentrate around true labels, indicating that often model almost correctly finds a boundary, having a shift by +1/-1 label only.

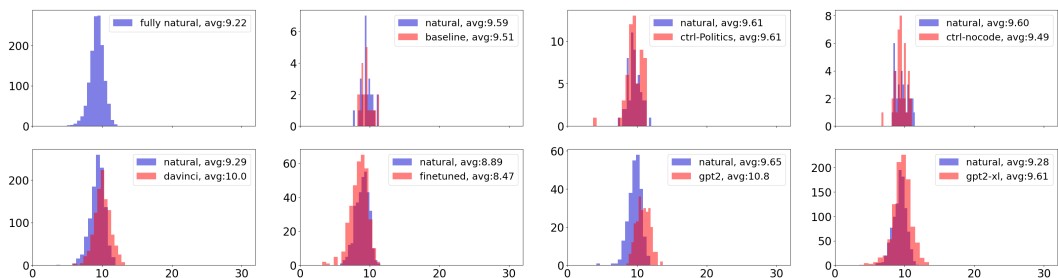

Figure 13: TLE dimension distributions for the sentences in the RoFT dataset by generator models

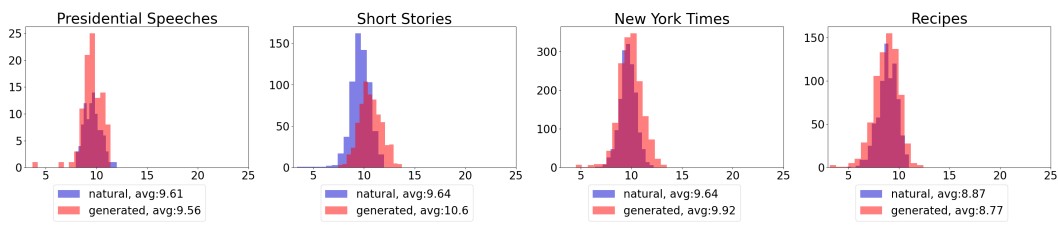

Figure 14: TLE dimension distributions for the sentences in the RoFT dataset by topics

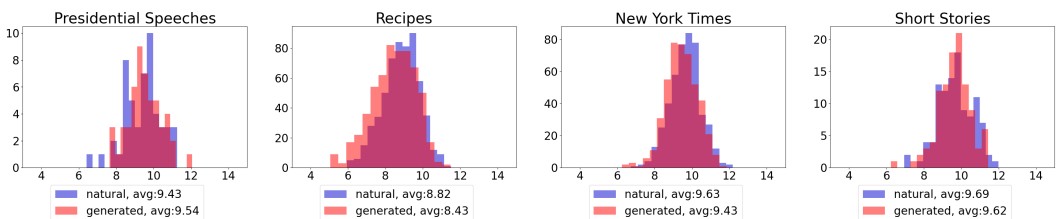

Figure 15: TLE dimension distributions for the sentences in the *RoFT-chatgpt* dataset by topics

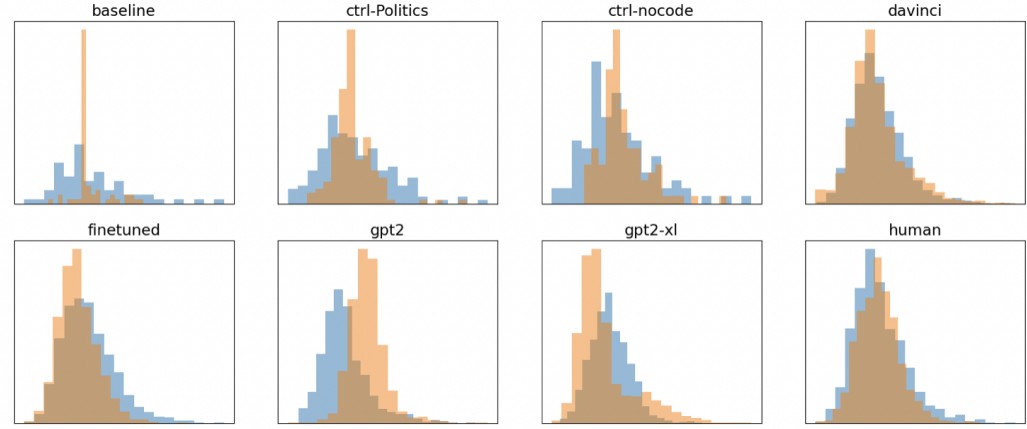

Figure 16: Perplexities of the last sentence in RoFT by model obtained from GPT-2; *blue* — distribution on the in-domain set, i.e., the entire dataset except the speficied generator; *orange* — on the out-of-domain set, i.e., data from the specified generator.

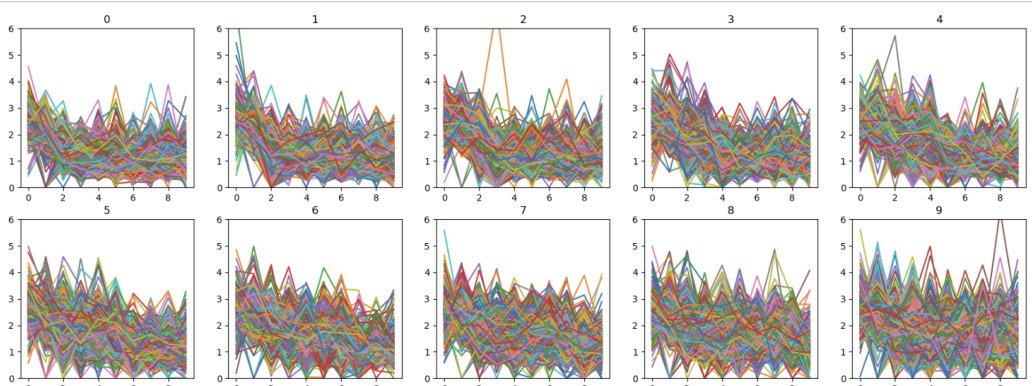

Figure 17: Sentence perplexities in the *RoFT-chatgpt* dataset by label obtained from Phi-1.5 model. X axis: sentence index in the text, Y axis: sentence perplexity.

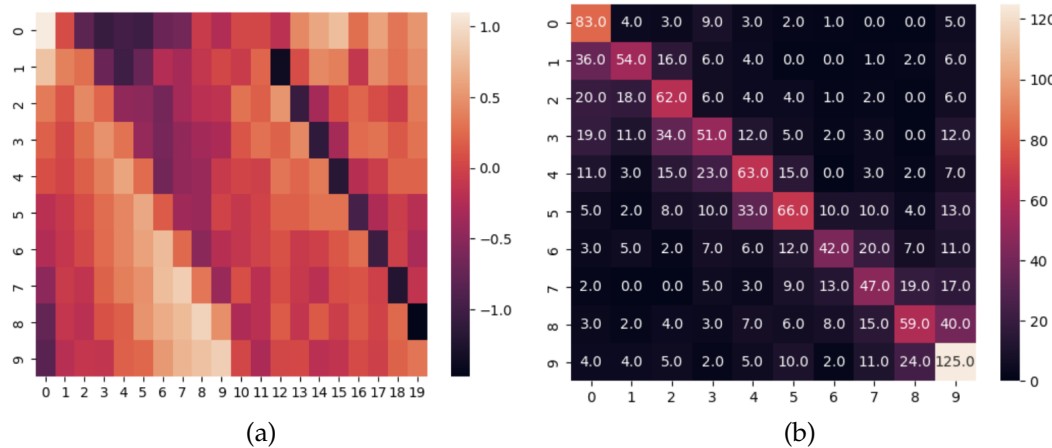

(a)                                    (b)

Figure 18: Analysis of the logistic regression trained on sentence perplexities on the *RoFT-chatgpt* dataset (*Perplexity + LogRegr* in the tables): (a) heatmap of the coefficients; (b) confusion matrix for test set predictions of logistic regression trained on sentence perplexities on the *RoFT-chatgpt* dataset.

## G   Examples of correctly classified and misclassified texts

We provided more examples of correctly and incorrectly classified texts at the Figures 19 and 20.

The girl sat, eyes wet.
"Get up.
Don't give up now," her friend said, placing a comforting hand on her shoulder.
The girl shook her head, tears streaming down her cheeks.
"I can't do it.
I'm just not strong enough.
" "You are," her friend insisted.
"Look at all the obstacles you've overcome already.
You're stronger than you think.
" The girl took a deep breath and wiped her tears.

Figure 19: Correctly classified sample input from the ROFT-chatgpt dataset colored according to the perplexities of Phi-1.5 on each sentence; "greener" text corresponds to higher mean perplexity, "blue-er", to higher standard deviation of the perplexity, red text shows where both mean and std are low. The prompt is above the dotted line, the rest is generated.

HOW TO MAKE: Canadian Bacon And Chicken Pasta Bake Ingredients: 1 (8 ounce) package ziti pasta nonstick cooking spray 1 lb chicken breast, diced 1 medium onion, diced 3 cloves garlic, minced 1 green pepper, chopped 2 zucchini, chopped 2 (15 ounce) cans tomatoes seasoned with basil garlic & oregano 1 (6 ounce) can tomato paste 14-12 teaspoon crushed red pepper flakes 1 (15 ounce) container fat-free ricotta cheese 1 (6 ounce) package Canadian bacon 1 cup light mozzarella cheese 2 tablespoons parmesan cheese.
Preheat oven to 350 degrees.
Cook pasta according to package directions without added fat or salt.
Drain and set aside.
In a large skillet coated with nonstick cooking spray over medium heat, cook chicken until browned and cooked through.
Add onion, garlic, green pepper, and zucchini.
Cook until vegetables are tender.
Stir in canned tomatoes, tomato paste, and crushed red pepper flakes.
Cook for 5 minutes.
In a large mixing bowl, mix together cooked pasta, ricotta cheese, and Canadian bacon.

"We do love you, son," my dad said.
"I know, Dad," I replied with a small smile.
"I love you too.
" It had been a long time since we had said those words to each other.
Our relationship had been strained for years, but we had recently started working on it.
It wasn't easy, but we were making progress.
"I just want you to know that no matter what happens in life, we'll always be here for you," my dad continued.
"We may not always agree with your decisions or understand what you're going through, but we'll always love you unconditionally.
" Tears welled up in my eyes as I hugged my dad.
It was a powerful moment, one that I would always cherish.

Figure 20: Misclassified inputs from the ROFT-chatgpt dataset colored according to the perplexities of Phi-1.5 on each sentence. At the first picture, generated text starts from the fourth sentence, but perplexity-based classifier predicts that it starts from the fifth sentence; at the second picture, generated text starts from the second sentence, but perplexity-based classifier predicts that it starts from the fifth sentence.

