# OpenReview forum: "AI-generated text boundary detection with RoFT"
_colmweb.org/COLM/2024/Conference — COLM_

### Official Review · Reviewer_Zghr · 2024-05-11

**Rating:** 5
**Confidence:** 4
**Ethics Flag:** 1

**Summary:**

This paper tackles the problem of detecting the boundary between human-written and machine-generated text in mixed texts, which is an increasingly common real-world scenario that has not been well-studied. The authors adapt several existing artificial text detection techniques to this boundary detection task and evaluate them on the RoFT (Real or Fake Text) dataset as well as a new RoFT-chatgpt dataset they created using GPT-3.5 generations.

**Reasons To Accept:**

1.	The study targets the detection of human-written and AI-generated text boundaries in mixed texts, a growing concern due to the prevalence of large language models. This problem has not received much attention in the literature, making this study highly relevant and timely.
2.	The researchers evaluated various adapted artificial text detection methods on the RoFT dataset and their new RoFT-chatgpt dataset. They tested the methods in different scenarios, including in-domain, cross-domain, and cross-model, offering valuable insights into each approach's strengths, weaknesses, and dataset property challenges.
3.	The paper adapts and assesses a range of techniques, establishing foundational baselines for the task. The authors' detailed result analysis and discussion of current limitations highlight promising avenues for future research.

**Reasons To Reject:**

1.	Technically, this paper mainly conducts benchmarking on the RoFT dataset, primarily using evaluation metrics. It is more like a technical report than a research paper with novel contributions.
2.	To my understanding, detecting the boundary between human-written and machine-generated parts should go beyond sentence classification. It would be better to localize which parts of the paragraph are likely written by a human and which parts of the article are likely written by AI. However, the current evaluation in this paper is doing binary classification for each sentence in the RoFT dataset, which may not capture the nuances of the task.
3.	Some parts of the paper are not clear. For example, in the evaluation section on artificial text boundary detection, the terms "SoftAcc" and "mean squared error" are not explicitly defined. It would be better to include mathematical notations to clarify these metrics.
4.	While the paper mentions the limitations of the methods in general terms, it lacks a deep dive into specific instances where the models failed. Providing concrete examples and analyzing the reasons behind the failures could offer valuable insights for future improvements in boundary detection techniques.

---

> ### Author Rebuttal · Authors · 2024-05-29
>
> Thank you for your review, and let us discuss your concerns.
>
> > To my understanding, detecting the boundary between human-written and machine-generated parts should go beyond sentence classification. It would be better to localize which parts of the paragraph are likely written by a human and which parts of the article are likely written by AI.
>
> We understand that the task of boundary detection by sentences differs from the most general task of localizing the parts of the paragraph that are likely to be written by AI. However, the general task is complicated and under-explored.  At the same time, most existing work focuses on solving the task in a much simpler setting of binary classification of whole texts into two classes, human- or AI-written. Thus, we find it reasonable to depart from the setting of binary classification of texts towards solving the most general task by first approaching a slightly relaxed problem of detecting AI-written parts of the text as a task of boundary detection by sentences. If we can solve this task well enough, we can move forward to more complex problem statements.
>
> > Some parts of the paper are not clear. For example, in the evaluation section on artificial text boundary detection, the terms "SoftAcc" and "mean squared error" are not explicitly defined. It would be better to include mathematical notations to clarify these metrics.
>
> Thank you for pointing this out. The paper mentions that SoftAcc is the percentage of predictions that differ from the correct label by at most one, but we agree that it would be better to define such terms in mathematical notation, and we will include the definitions in the camera-ready version of the paper if accepted.
>
> > While the paper mentions the limitations of the methods in general terms, it lacks a deep dive into specific instances where the models failed. Providing concrete examples and analyzing the reasons behind the failures could offer valuable insights for future improvements in boundary detection techniques.
>
> We fully agree that qualitative analysis of specific examples would benefit the paper and make it more clear where current modes fail and perhaps provide intuition for potential improvement. The paper discusses some particular data properties that complicate cross-domain boundary detection in the “Data Properties” section (p. 8), and we will be able to add more qualitative analysis to the camera-ready if accepted since an extra page will be allowed in the paper.

---

> > ### Comment · Reviewer_Zghr · 2024-06-03
> >
> > Thanks for your response. I tend to maintain my original scores.

---

> ### Comment · Area_Chair_44QH · 2024-06-03
> **Please respond to the authors rebuttal**
>
> Hello reviewer,
> As a reminder, the discussion period ends on Thursday, June 6. Please take a look at the author's rebuttal and acknowledge if they have adequately addressed the reasons you gave to reject.
> Thanks,
> AC

---

> > ### Author Response · Authors · 2024-06-04
> > **Response to Area Chair 44QH and Reviewer Zghr**
> >
> > Dear Area Chair 44QH, dear Reviewer Zghr, we are grateful to the reviewer for the constructive feedback, we believe that we have addressed most of the reviewer's concerns, adding further improvements to the paper. We respectfully disagree with the  «technical report» remark. The distinction between a research paper and a «technical report» is rather subjective,
> > our paper contains new ideas, novel experiments, and analysis, that we believe offer useful insights to the community.

---

### Official Review · Reviewer_LWRU · 2024-05-11

**Rating:** 8
**Confidence:** 3
**Ethics Flag:** 1

**Summary:**

The paper titled investigates the challenge of detecting boundaries between human-written and machine-generated text, an issue increasingly relevant due to the proliferation of LLMs. Utilizing the real or fake text dataset, which comprises texts with both human-written and machine-generated sections, the study tests several methodologies adapted for this complex detection scenario, moving beyond traditional binary classifications. Key approaches include perplexity-based methods that evaluate text unpredictability, adaptations of classifiers like RoBERTa that leverage text embeddings to detect shifts in text generation sources, and innovative applications of topological data analysis to determine the intrinsic dimensionality of text data, which provides robustness against domain and model shifts. The results suggest that PPL based methods and intrinsic dimension estimation offer promising results, outperforming traditional binary classifiers and showing better adaptation to the nuanced challenges of environments with mixed human and AI-generated text. This research contributes valuable insights into artificial text detection, highlighting the potential for further exploration and refinement of these techniques to improve our ability to discern human from machine-generated content effectively.

**Reasons To Accept:**

1. The problem is very new, and any advancement is appreciated.
2. The authors have also advanced the dataset and released it for future research, given that the problem is very important in current times.
3. The paper showcases diverse experimental scenarios, such as cross-model and cross-generation comparisons, using both binary classification and perplexity-based evaluation.
4. An appropriate explanation of poor generalization scenarios is provided, which is not common to see in other papers.
5. The paper is very well written and explained, with high-quality visuals.
6. Three models—GPT, Phi, and LLaMA—are explored, which makes it a strong and robust framework.

**Reasons To Reject:**

1. Analysis is limited to only one datasets

---

> ### Author Rebuttal · Authors · 2024-05-29
>
> Thank you for your review and for your high evaluation of our work. Let us address your concern:
>
> > Analysis is limited to only one datasets
>
> Please see the answer about the one dataset above. This is indeed a novel problem setting, and at the time of performing experiments for the paper we were aware of only one preexisting dataset.

---

> > ### Comment · Reviewer_LWRU · 2024-06-05
> > **Maintaining original rating**
> >
> > Thank you for your comment to the review! I tend to maintain my original rating. Thanks

---

### Official Review · Reviewer_quBQ · 2024-05-11

**Rating:** 10
**Confidence:** 3
**Ethics Flag:** 1

**Summary:**

The paper examines a novel and interesting problem of detecting the transition between human-written and AI-generated text within documents that contain both. It uses updated datasets like RoFT and RoFT-chatgpt to test various detection methods, including RoBERTa-based classifiers, perplexity-based methods, and topological features. The findings reveal that while RoBERTa tends to overfit and struggle with generalization, perplexity-based methods offer a good balance of accuracy and generalization, especially when trained on synthetic data. Topological methods show promise in robustness across different domains but occasionally lack high accuracy. The study highlights significant challenges in cross-domain and cross-model generalization, particularly with short texts due to limited context and longer texts due to their complexity.

**Questions To Authors:**

Why was the choice of datasets limited to only one? Is there a specific reason pertaining to the problem being solved?

**Reasons To Accept:**

The paper aims to study a novel and upcoming challenge which is important study due to the wide application of LLMs in the domain of Natural Language Processing. This is particularly important in certain domains where authorship attribution is important.
This is a comprehensive study that uses a diverse array of methods, including RoBERTa-based classifiers, perplexity-based classifiers, and topological features. The authors perform a thorough analysis of the strengths and weaknesses of each method under different conditions including the characteristics of the dataset itself leading to certain performance trends. The analysis is done for real-time applications including cross-domain and cross-modal scenarios i.e., the paper discusses the practical implications of each tested method, providing a clear understanding of their applicability in real-world settings. The limitations of different approaches are clearly mentioned.

**Reasons To Reject:**

More explanations about the choice of parameter values may be helpful to understand the underlying methodology that has been adopted. For examples, the authors use a window size of 20 with strides of 5 when performing Topological Data Analysis (TDA). More explanation for the choice of 20 and 5 can be helpful. Are those values standard or dependent on text length and identified empirically?

Missing references can be corrected. For example, On page 5, the reference figure is missing in the text "(SoftAcc1; indeed, Fig. ?? in Appendix F shows that many misclassifications point to an adjacent class, which may be acceptable in real world applications)".

---

> ### Author Rebuttal · Authors · 2024-05-29
>
> Thank you for your review and for your high evaluation of our work.
>
> > More explanations about the choice of parameter values may be helpful to understand the underlying methodology that has been adopted. For example, the authors use a window size of 20 with strides of 5 when performing Topological Data Analysis (TDA). More explanation for the choice of 20 and 5 can be helpful. Are those values standard or dependent on text length and identified empirically?
>
> Thank you for the feedback, we add more clarifications on this in the final version of our paper.  Here are the answers to your specific questions:
> - Window size 20 was taken directly from “Intrinsic Dimensionality Estimation within Tight Localities: A Theoretical and Experimental Analysis” by Laurent Amsaleg et al., as the authors claim that their algorithm is stable enough “for `tight' localities consisting of as few as 20 sample points”.
> - Stride size 5 was chosen after the preliminary experiments. Initially, we tried smaller stride sizes, but we found out that in these cases the series became too long for the Time Series SVM algorithm, and it wasn't able to converge.
>
> > Missing references can be corrected. For example, On page 5, the reference figure is missing in the text "(SoftAcc1; indeed, Fig. ?? in Appendix F shows that many misclassifications point to an adjacent class, which may be acceptable in real-world applications)".
>
> Thank you for this remark, SoftAcc is the percentage of predictions that differ from the correct label by at most one; broken figure reference referred to Figure 18b. We will recheck the paper for missing references and typos before submitting the camera-ready version.
>
> > Why was the choice of datasets limited to only one? Is there a specific reason pertaining to the problem being solved?
>
> Yes, there was essentially only one dataset that met the following criteria:
> - it was labeled for boundary detection instead of just classifying texts as real or AI-generated;
> - it included human results, meaning experiments where humans labeled the boundaries;
> - it was available at the time of writing the paper, so that we could compare our results with previous work.
>
> Recently, another dataset has appeared, namely subtask C of Task 8 of a recent SemEval workshop: https://github.com/mbzuai-nlp/SemEval2024-task8 . However, the full labels of its test set (i.e. with the information about generating models) were only released after the CoLM deadline.

---

> > ### Comment · Reviewer_quBQ · 2024-06-05
> > **More clarity on the choice of empirical values**
> >
> > The response for choice of one dataset is reasonable and so is the mention about the window size of 20. Can you shed more light on the range of values selected for stride size. You did mention about the reason for rejecting smaller stride values but it could shed more light on your choice if you can mention the range of values initially considered for the preliminary experiments and also mention what did you observe for not only the smaller stride values but also the larger ones.

---

> > > ### Author Response · Authors · 2024-06-05
> > > **Stride values**
> > >
> > > In our preliminary experiments, we tried only stride values 1, 2, and 5. We didn't try values above 5.

---

### Decision · Program_Chairs · 2024-07-10

**Decision:**

Accept

**Comment:**

This paper examine which methods are most effective for solving the boundary detection problem---that is, identifying when text transitions from machine-generated to human-written. The reviewers emphasize that this is a relatively new and important problem for which there is little prior work. They also note that the paper's analysis is thorough and well-written.